# A Comparative Study of the Adherent-Invasive *Escherichia coli* Population and Gut Microbiota of Healthy Vegans versus Omnivores

**DOI:** 10.3390/microorganisms8081165

**Published:** 2020-07-31

**Authors:** Rebecca Veca, Christian O’Dea, Jarred Burke, Eva Hatje, Anna Kuballa, Mohammad Katouli

**Affiliations:** School of Health and Sport Sciences, University of the Sunshine Coast, Maroochydore DC, QLD 4558, Australia; rebecca.veca@gmail.com (R.V.); cao003@student.usc.edu.au (C.O.); jrb040@student.usc.edu.au (J.B.); ehatje@usc.edu.au (E.H.); akuballa@usc.edu.au (A.K.)

**Keywords:** AIEC, vegan, omnivore, diet, gut microbiota

## Abstract

Adherent-invasive *Escherichia coli* (AIEC) strains carry virulence genes (VGs) which are rarely found in strains other than *E. coli*. These strains are abundantly found in gut mucosa of patients with inflammatory bowel disease (IBD); however, it is not clear whether their prevalence in the gut is affected by the diet of the individual. Therefore, in this study, we compared the population structure of *E. coli* and the prevalence of AIEC as well as the composition of gut microbiota in fecal samples of healthy participants (*n* = 61) on either a vegan (*n* = 34) or omnivore (*n* = 27) diet to determine whether diet is associated with the presence of AIEC. From each participant, 28 colonies of *E. coli* were typed using Random Amplified Polymorphic DNA (RAPD)–PCR. A representative of each common type within an individual was tested for the presence of six AIEC-associated VGs. Whole genomic DNA of the gut microbiota was also analyzed for its diversity profiles, utilizing the V5-V6 region of the16S rRNA gene sequence. There were no significant differences in the abundance and diversity of *E. coli* between the two diet groups. The occurrence of AIEC-associated VGs was also similar among the two groups. However, the diversity of fecal microbiota in vegans was generally higher than omnivores, with *Prevotella* and *Bacteroides* dominant in both groups. Whilst 88 microbial taxa were present in both diet groups, 28 taxa were unique to vegans, compared to seven unique taxa in the omnivores. Our results indicate that a vegan diet may not affect the number and diversity of *E. coli* populations and AIEC prevalence compared to omnivores. The dominance of *Prevotella* and Bacteroides among omnivores might be accounted for the effect of diet in these groups.

## 1. Introduction

Inflammatory bowel disease (IBD) represents a group of spontaneous, chronic inflammatory conditions of the gastrointestinal (GI) tract, including Crohn’s disease (CD), ulcerative colitis (UC), and, prospectively, colorectal cancer (CRC) [1]. The global financial and health-associated burden of IBD impacts approximately five million people per year, with the highest incidence in developed countries and steadily increasing in developing countries [2]. Although the exact etiology of IBD is yet to be fully elucidated, a dysregulated immune response, host genetics, environmental influences, and gut microbial dysbiosis are widely established factors in IBD pathogenesis [3]. Recent studies indicate that mucosa-associated *Escherichia coli* are found in higher abundance in people with CD, CRC, and, to a lesser extent, UC [4]. These strains, termed adherent-invasive *E coli* (AIEC), typically diffusely adhere (DA) to and invade intestinal epithelial cells in culture as well as survive and replicate within macrophages [5].

AIECs have been associated with specific virulence genes (VGs) that are not found, or rarely found, in other strains of *E. coli* [4]. These include outer membrane protein C (*omp*C) and afimbrial adhesin C (*afa*C) that mediate diffuse adhesion to and invasion of GI epithelial cells and induce vascular endothelial growth factor expression, which contributes to angiogenesis and tumor development [4,6]. Long polar fimbriae A gene (*lpf*A) enables the bacterium to translocate across the follicle-associated epithelium of the Peyer’s patches via microfold cells (M-cells) [4]. High temperature requirement A gene (*htr*A) and oxidoreductase disulphide bond A protein (*dsb*A) support survival and replication of AIEC in macrophages [4], and, finally, colibactin-A gene (*clb*A), which is located on the polyketide synthase pathogenicity island, is a genotoxin that causes damage to the DNA of enterocytes, leading to carcinoma development [7]. While AIEC strains are implicated in the pathogenesis of IBD and CRC, they have also been found in lower abundance in healthy individuals, animals, and the environment [8]. The presence of these bacteria in the GI tracts of healthy individuals may therefore suggest a risk for development of IBD in susceptible hosts [9].

Diet is one of the factors known to alter the gut microbiota and is proposed to influence dysbiosis in susceptible individuals and IBD pathogenesis [10]. Schreiner et al., 2018 found a significant difference in the microbial composition of vegan IBD patients and omnivorous IBD patients [11]. This could probably be due to the fact that vegan diets exclude meat, poultry and fish, dairy, eggs, gelatin, and, in some instances, honey [12]. Differences in the gut microbial composition are also reported in healthy individuals consuming a vegan versus an omnivorous diet [13]. Importantly, a large cohort study found a significantly lower abundance of *E. coli* and *Enterobacteriaceae* in participants on a vegan diet compared to an omnivorous diet [14]. This is consistent with results from two other vegetarian versus omnivore studies [15,16]. However, other vegetarian studies presented no change in the abundances of these bacteria [17], which may be due to the different sampling methods and population sizes used in these studies. Other differences in microbial composition between vegans and omnivores have also been observed for *Bacteroides* and *Prevotella* abundance. Consistent amongst these studies, *Bacteroides* are found in significantly higher abundance in omnivores than vegans; however, results are inconsistent for *Prevotella*, which are either increased, decreased, or unchanged in separate studies [14,18,19,20].

The relationship between diet, gut microbiota, and IBD pathogenesis suggests that the presence of AIEC might differ for different diet types. To the best of our knowledge, the identification of AIEC in healthy individuals has not yet been investigated for vegan and omnivorous diets. Considering the altered microbial composition resulting from a vegan diet, we hypothesized that the *E. coli* diversity, as well as the total diversity of microbiota, could have the potential to vary between vegans and omnivores. We also postulated that the prevalence of AIEC might be reduced in a vegan diet as a result of this. The aim of this study was therefore to compare the population structure and presence of virulence genes of the *E. coli* populations as well as the diversity profile of gut microbiota in healthy individuals consuming a vegan diet versus an omnivore diet. Special attention was given to the presence of AIEC strains in these two groups.

## 2. Materials and Methods

### 2.1. Participants

A total of 61 participants were recruited from South-East Queensland, Australia, between January and May of 2018. Participants self-identified as consuming either a vegan diet or an omnivorous diet (Table 1). A vegan diet was identified as the exclusion of meat, poultry, fish, dairy, eggs, and gelatin, whereas an omnivorous diet included these products. These dietary characteristics are well established for a vegan and omnivorous diet throughout the general population and scientific community [12]. A minimum of 4 weeks’ consumption of a vegan diet prior to sample collection was used as the criterion for this study, as altered prevalence of bacterial species is reported after this time [21]. Healthy participants were recruited via survey, face-to-face invitation, and self-assessment. They also confirmed no prior use of antibiotics or probiotics 3 months prior to undertaking the study. This timeframe was used for the study due to the impact of antibiotics and probiotics on the gut microbiota [22,23]. A total of 7 participants (6 omnivores, 1 vegan) identified as smokers, and participants were considered apparently healthy, determined by the absence of health conditions associated with the GI tract upon involvement in the study. This definition was considered the most relevant for this study due to the wide variability in a healthy status [24]. The length of time for which participants in the vegan group had consumed a vegan diet varied: 1–10 months (*n* = 5), 11–20 months (*n* = 11), 21–30 months (*n* = 4), and more than 31 months (*n* = 13). A consent form and questionnaire were provided to all participants to accumulate these data. Human ethics for this study was approved by the University of the Sunshine Coast (S/17/1122).

### 2.2. Fecal Sample Collection

Fecal samples were self-collected by participants in a sterile container provided in the fecal collection kit supplied to each participant. Fecal samples were collected in sterile containers and placed in a sealed plastic sleeve, within a sealed envelope, and stored at 4 °C, and they were transferred to the laboratory on ice on the day of sampling. Fecal samples were processed for the cultivation and isolation of *E. coli*, as described earlier [25], and for DNA extraction of the whole microbial genome for analysis of microbial diversity within 6 h of collection, as previously described [26].

### 2.3. Enumeration of E. coli Strains and DNA Extraction

Enumeration of *E. coli* was performed as described previously [14]. Briefly, fecal samples were 10-fold serially diluted using phosphate buffered saline (PBS, pH 7.3) and cultivated on MacConkey agar no. 3 (Oxoid, Melbourne, Australia). After overnight incubation at 37 °C, the number of *E. coli* isolates were calculated as CFU/g of feces. From each sample, 28 *E. coli*-like colonies were randomly selected and subjected to *E. coli* confirmatory tests using PCR amplification of the universal stress protein (*usp*A) gene with species-specific primers (F 5′-CCG ATA CGC TGC CAA TCA GT-3′ and R 5′-ACG CAG ACC GTA GGC CAG AT-3′) (Invitrogen, Melbourne, Australia), as previously described [27]. Confirmed *E. coli* isolates were saved in tryptone soya broth (TSB; Oxoid, Melbourne, Australia) containing 20% (*v/v*) glycerol and stored at -80 °C for further analysis.

Chromosomal DNA of *E. coli* strains grown on nutrient agar (NA; Oxoid, Melbourne, Australia) was extracted using the traditional boiling method [28]. In brief, individual colonies from the NA agar were suspended in 150 µL of Tris-EDTA (TE) buffer and boiled for 15 min on a dry heat block at 100 °C (Ratek, Victoria, Australia). Suspensions were centrifuged at 13,500 rpm for 10 min and the final supernatant DNA was stored at -20 °C for further analysis.

### 2.4. Clonality Analysis of E. coli Strains

All strains were typed for their genetic relatedness using the randomly amplified polymorphic DNA (RAPD) PCR method with the primer PB1 (5′-GCGCTGGCTCAG-3′) (Invitrogen, Melbourne, Australia), as described previously [29]. Molecular weight markers (100 bp and 1 kb) were used in all experiments. All PCR experiments for this study were performed in a Mastercycler gradient (Eppendorf, New South Wales, Australia) thermal cycler. Primers and PCR conditions are given in Appendix A. Gel electrophoresis was performed using a 1 kb ladder (Axygen Scientific, Cambridge, Australia), with a 1.5% agarose gel in 0.6× Tris-borate-EDTA (TBE) buffer, run at 90 V for 3.5 h, adapted from a previous method [30]. Images were viewed and photographed under UV and imported to GelCompar II version 6.5 (Applied Math, Sint-Martens-Latem, Belgium) for analysis. As in a previous study [30], clustering analysis was performed with 0.5% optimization and 2% tolerance using the unweighted pair group method with arithmetic mean UPGMA method with Dice’s coefficient. Strains that clustered above 95% were considered to belong to the same clonal type (CT), while strains that presented visually different banding patterns to the identified CTs were termed a single type (ST) (Appendix B, Figure A1). A representative strain of each CT was randomly selected for further analysis. To ensure that the selected strains from each CT were truly representative of all strains in that CT, strains from three CTs (three from each) were analyzed to ensure the reproducibility of the results. The diversity of *E. coli* strains in each participant was measured as Simpson’s diversity index (Di) [31].

### 2.5. Detection of AIEC Associated Virulence Genes

Strains representing different CTs were tested for the presence of six AIEC-associated VGs, i.e., *afa*C, *lpf*A, *htr*A, *dsb*A, *clb*A and *omp*C, using previously described protocols [4]. Previously confirmed positive laboratory strains AAA-172, KIC-2, KIC-2, RBH-128, RBH-128, and KIC-2 were used as positive controls for each of the above VGs, respectively, and sterile filtered Milli-Q water was used as the negative control. The PCR conditions for *afa*C, *lfp*A, and *clb*A genes included 16.8 µL of sterile filtered Milli-Q water, 2.5 µL of 10× reaction buffer (Bioline, Sydney, Australia), 1 µL of 10 mM dNTPs (ThermoFisher, Melbourne, Australia), 1 µL of 50 mM MgCl2 (Bioline, Sydney, Australia), 0.75 µL of 10 µM forward and reverse primers (Invitrogen, Melbourne, Australia), 0.2 µL of 5 U/µL Biotaq DNA polymerase (Bioline, Sydney, Australia), and 2 µL of DNA template to create 25 µL total volume. PCR conditions were identical for the other genes, except 16.5 µL of water and 1.5 µL of MgCl2 for *htr*A and *omp*C, and 15.8 µL of water and 2 µL of MgCl2 for dsbA. Gel electrophoresis for all genes involved a 100 bp ladder (Axygen, Cambridge, Australia) and a 1.5% agarose gel in 0.6× TBE buffer run at 100 V for 1 h, and the gel images were viewed and photographed as described above.

### 2.6. Diversity Profiling of Fecal Microbiota

Whole genomic DNA was extracted from fresh fecal samples using the QIAamp PowerFecal DNA Isolation kit (Qiagen, Hilden, Germany), as per the manufacturer’s protocols. This method has been shown to yield sufficient concentration and quality of DNA for use in 16S rRNA amplicon analysis [32]. For logistical reasons, 26 fecal samples (18 samples from the vegan group and 8 samples from omnivore group) were pooled to create a total of 5 vegan and 5 omnivore composite samples. DNA samples were submitted to the University of Minnesota Genomics Center (UMGC) for 16S rRNA sequencing. Pipeline analysis and associated primers, reagents, and equipment were conducted using previously described methods to reduce potential source library bias [33]. Primers used were 784F (5′-RGGATTAGATACCC-3′) and 1064R (5′-CGACRRCCATGCANCACCT-3′) to amplify regions V5-V6 of the 16S rRNA sequence with KAPA HiFidelity Hot Start Polymerase. Samples were analyzed with initial cycling conditions of 95 °C for 5 min, with 25 cycles of 98 °C for 20 s, 55 °C for 15 s, and 72 °C for 1 min; followed by the addition of Illumina adapter, barcode sequences and dual index PCR for 10 cycles [33]. Sequences were purified by gel, pooled, and pair-end sequenced (300 nt read length) using the Illumina MiSeq platform (Illumnia, Inc., San Diego, CA, USA) [33].

### 2.7. Bioinformatics Analysis

Demultiplexed 16s rRNA forward and reverse reads were analyzed using the Quantitative Insights Into Microbial Ecology (QIIME) 2 pipeline software (v. 1.9.1) [34]. Already demultiplexed human pooled sequences were subjected first to joining forward and reverse reads using artifact “Paired End Sequences with Quality” (QIIME 2). The joined reads were denoised and filtered of erroneous reads which compromised the final output. The “Dada2 denoise-paired” artifact trimmed forward, and reverse primer ends of the V5-V6 amplicon region and truncated the 170bp region from the sequence reads. Noisy reads and repetition were removed within the resultant dataset. Resultant output .qza files underwent sequence alignment, mask/filter alignment, and production of phylogenetic tree files. Biases were removed using “diversity alpha–rarefaction” at a maximum depth of 118143 to accommodate and filter only quality reads [34]. Finally, taxonomic assignment was undertaken with open reference database, SILVA-132-99 classifier, one of the reference datasets used by QIIME 2 for operational taxonomic unit (OTU) picking. SILVA software was utilized over other databases for OTU assignment due to recent database updates [35].

Classified taxa were tabulated with a custom metadata file pertaining to the human samples’ volunteer information that was provided, i.e., gender, diet, age, location, smoking status. Exported taxonomy files were converted to BIOM format for subsequent output analysis into open source metagenomic data facilitation software, CALYPSO [36]. The microbial richness and diversity of the vegan and omnivore samples were determined by total sum scaling (TSS) abundance normalization. The diversity of the samples was calculated by Shannon’s diversity index. Distances between microbial communities in the vegan and omnivore groups were calculated and visualized by Principal Coordinate Analysis (PCoA). Venn diagram formats were utilized to visualize the amount of unique and core microbial communities between the vegan and omnivore groups. All post data for this analysis were visualized using CALYPSO [36] and were considered statistically significant when *p* < 0.05 using a Fisher’s exact test.

### 2.8. Statistical Analysis

Statistical analysis was performed using GraphPad Prism v. 8 for Windows (GraphPad Software, San Diego, CA, USC). An unpaired nonparametric Mann–Whitney test was used to compare the significance of difference between the number of *E. coli* as well as their diversity and the results were expressed as mean ± standard error of mean between the vegan and omnivore groups. A nonparametric one-way analysis of variance (ANOVA), the Kruskal–Wallis test, was used to test the significance of differences between the prevalence of AIEC-associated VGs between both groups. The relationships between the prevalence of AIEC and other factors of interest for the vegan and omnivore groups were evaluated using Fisher’s exact test with Pearson’s correlation coefficient. Differences were considered statistically significant when *p* < 0.05.

## 3. Results

### 3.1. Abundance and Diversity of E. coli

No significant differences were found in the number of *E. coli* isolates per gram of feces and their diversity between the vegan and omnivore groups (Table 2). The mean number of *E. coli* isolates per gram of feces was in the range of 3.4 × 10^3^–7.6 × 10^7^ for vegans and 3.4 × 10^3^–4.7 × 10^6^ for omnivores (Table 2). *E. coli* diversity in both groups was generally low and ranged from 0 to 0.860 for vegans and from 0 to 0.717 for omnivores, a result that indicates the dominance of one major group of *E. coli* isolates in each individual (Table 2). The abundance and diversity of fecal *E. coli* in male vegan participants were generally higher than male omnivore participants, but these figures were not statistically significant (Table 2).

Among 1708 *E. coli* isolates typed using RAPD-PCR, a total of 1660 colonies belonged to 99 unique common types (CTs) and 48 were single types (STs) (Table 3). Of these, 59 CTs and 30 STs were found among vegans, while 40 CTs and 18 STs were found among omnivores. To test whether the randomly selected isolates from each CT were true representations of that CT, we randomly tested three isolates from three CTs (three isolates from each CT) and tested them for the presence of VGs. The results indicated that all isolates within each CT carried identical VG profiles (Appendix A).

### 3.2. Prevalence of AIEC Associated VGs

All 99 CTs were tested for the prevalence of six AIEC-associated VGs. The distribution of these VGs among the CT varied among different CTs, with one CT, i.e., CT8, possessing all six VGs. In all, 61 CTs (62%) carried more than four out of the six VGs tested (Appendix A). The occurrence of these genes within each CT, however, did not differ significantly between vegans and omnivores (Table 4).

### 3.3. Diversity of the Gut Microbiota

Whilst the abundance of *E. coli* between the vegans and omnivores did not differ significantly, there was a significantly higher abundance of *Enterobacteriaceae* among vegans (*p* < 0.001) (Figure 1a). Of the five composite samples tested from each group, one composite sample belonging to the omnivore group, i.e., O2, had an exceptionally high abundance of *E. coli* compared to other samples from omnivores that showed a very low value (Figure 1b).

Diversity profiling of the gut microbiota using composite samples showed a difference in the composition and/or abundance of different groups of bacteria in vegans and omnivores (Figure 2a). *Prevotella* and, to a lesser extent, Bacteroides constituted the dominant taxa among both groups. Data were visualized using the top 20 genera and 1000 OTUs (Figure 2). For further information on the top 100 genera using 5000 OTUs, see Appendix A. No relationships were found between age, gender, and months spent consuming a vegan diet (Figure 2b).

A PCoA analysis was conducted based on the diversity of gut microbiota to establish the overall similarity distance of the microbiota of vegans and omnivores. The results showed two distinct clusters of gut microbiota within the two diet groups, with one omnivore sample showing close similarity to the vegan cluster (Figure 3).

The diversity of microbiota in vegans was generally higher and more homogenous than omnivores (Figure 4a). The mean diversity of vegan microbiota (3.42 ± 0.08, ranging from 3.2 to 3.7) was slightly higher than that of omnivores (2.96 ± 0.3), which ranged from 2.0 to 3.9. This could be partly due to the higher number of taxa identified in the former group. Of the 123 taxa identified, 88 were shared between the two groups, with vegan and omnivore microbiota composed of 28 and seven unique taxa, respectively (Figure 4b) (Appendix B, Table A1).

## 4. Discussion

This study demonstrated a snapshot of the influence of vegan and omnivore diets on the abundance of *E. coli* and the diversity of gut microbiota in healthy individuals. Consistent with our hypothesis, the diversity of the gut microbiota in vegans was generally higher compared to omnivores, though not significant. This was partly due to the higher number of unique taxa (i.e., 28) identified among vegans’ gut microbiota as opposed to seven unique taxa found among omnivores. Despite this difference in diversity, the number of *E. coli* isolates and their diversity did not differ significantly between the two diet groups. Typically, *E. coli* isolates comprised only 0.5% of the gut microbiota despite their involvement of several intestinal and extra-intestinal infections [37]. The general *E. coli* population from feces in our study was relatively low, with abundance only slightly higher in male vegans than male omnivores. To the best of our knowledge, higher *E. coli* abundance has been observed for males versus females [38], but not for males of different diet types. Similarly, very little work has been done to compare the abundance of *E. coli* between the vegan diet and other diet groups.

Fecal *E. coli* isolates in healthy adults are shown to be dominated by one or two major types [39]. This was consistent with our study, as 99 CTs that were found among the whole sample size (*n* = 61) comprised 97% of *E. coli* colonies tested. This translates to approximately 1.5 representative strains of *E. coli* in omnivores and 1.7 in vegans per individual. However, an interesting finding in our study was that all CTs were found to be unique types. Considering that *E. coli* is found abundantly in the environment and can be circulated among human hosts [8], we expected to find some similar CTs within the few individuals. This might be explained partly by the small sample size of our study and/or by the interindividual variability among resident strains and hosts, as seen in other studies [40,41].

Zimmer et al. (2012) compared a vegan and omnivore diet and found significantly reduced *E. coli* abundance in vegans [14], which could be due to their higher sample size compared to our study, implying that our results might be different with a higher sample number. Alternatively, this difference could be due to the difference in geographical and environmental factors. A study comparing the microbiota of vegetarians and omnivores has found a similar abundance of *E. coli* to our study [16], whilst another study found decreased *E. coli* abundance in vegetarians [15]. Hence, there is some inconsistency regarding the abundance of *E. coli* in different diets including the vegan diet. In the present study, we also did not find any difference in the prevalence of AIEC-associated VGs between the vegan and omnivore groups except one CT (representing eight *E. coli* strains) that carried all six VGs in a 57-year-old male who had consumed a vegan diet for 12 months. Although AIECs are less abundant in healthy individuals compared to IBD patients [4], our results indicated that AIEC-associated VGs can be found in over half of the dominant *E. coli* strains in healthy populations. These results are consistent with Rahmouni et al. (2018), who proposed high carriage of AIECs in healthy individuals and wildlife [8].

It has to be noted that our study only tested CTs representing dominant *E. coli* and not STs. Owrangi et al. (2017) showed that the carriage of AIEC-associated VGs of non-dominant strains were significantly higher than dominant strains. This suggests that the prevalence of AIEC might be even higher if STs were also tested for VGs. Considering the above, our data support the notion that AIEC might be considered a pathobiont. It has to be noted that, like any other VGs, AIEC-associated VGs might also be exchanged among bacterial strains within the GI tract [42] and therefore the dominant strains could have received these VGs from non-dominant or transient strains.

In our study, the abundance of *E. coli* populations as well as the gut microbial profiles of the vegan and omnivore groups were compared. The general diversity of microbiota was higher for vegans than for omnivores; however, this was not statistically significant and did not affect *E. coli* abundance. Losasso et al. (2018) found a higher number of unique bacterial taxa in vegans than omnivores [20]. A similar result was also found in our study, in which vegans’ gut microbiota carried over 16% more unique bacterial taxa than omnivores. Although diet can alter the gut microbiota, the dominant and non-dominant species are variable among individuals [40]; therefore, unique taxa cannot be attributed to diet alone.

Two dominant genera—i.e., *Bacteroides* and *Prevotella*—were found among vegans and omnivores. Typically, *Bacteroides* are associated with long-term diets containing high protein and animal fats [43]. This supports the increased abundance of *Bacteroides* in the omnivore group than the vegan group, although this figure was not statistically significant, which was consistent with the results of a previous study [20]. *Prevotella*, on the other hand, are typically associated with diets that are high in carbohydrates and fiber [43], although different abundances of fecal *Prevotella* subspecies have been found in omnivores and in individuals consuming plant-based diets [44]. This might explain the dominant abundance of *Prevotella* in both vegans and omnivores.

Our study also showed a statistically significant increase in *Enterobacteriaceae* in vegans compared to omnivores. This result was interesting, as a previous study found a statistically significant decrease in this group of bacteria in vegans [14]. Other studies comparing vegetarians and omnivores also found a decreased abundance of *Enterobacteriaceae* in vegetarians [15,16], while another study found no differences between these two groups [17]. Although exact dietary intakes within the vegan and omnivore groups were not recorded for this study, one possible explanation for our findings is the high fiber fermentable content of flaxseed and buckwheat in vegans [45]. Vegans are proposed to have a diet higher in carbohydrates, potentially buckwheat and flaxseeds, which are a recommended animal-free source of omega-3 fatty acids for the vegan diet [46,47]. This notion was proposed by Pulkarbek et al. (2017), who suggested that increased consumption of flax and buckwheat is associated with increased levels of *Enterobacteriaceae* through fermentation, resulting in higher proliferation in feces [45]. Due to logistical reasons, the samples used for microbial profiling were analyzed as composite samples, and therefore the data do not represent the gut microbiota of individuals. Except for dietary exclusions in vegans, the participants in this study were not required to present a food journal or follow any strict dietary guidelines within the designated diet groups.

## 5. Conclusions

In conclusion, the abundance and diversity of *E. coli* populations as well as the prevalence of AIEC VGs in these strains did not differ between the two diet groups. Microbiota analysis revealed more unique taxa amongst vegans compared to omnivores. Specifically, *Bacteroides* was found in a higher proportion in the omnivore group, which could be due to the diet, whereas *Enterobacteriaceae* was higher in the vegan group. Further study is needed to identify the importance of these two groups in relation to the diet.

## Figures and Tables

**Figure 1 microorganisms-08-01165-f001:**
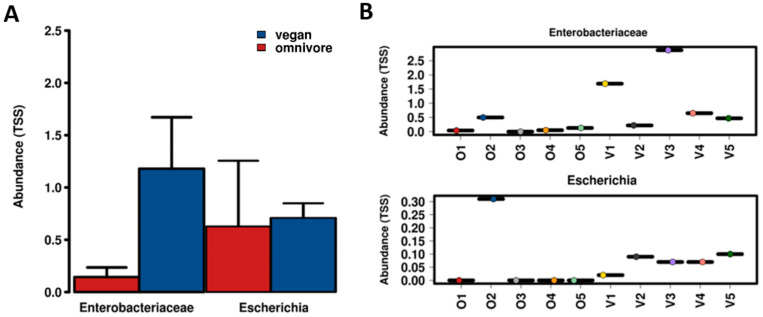
**A**: Abundances of *Enterobacteriaceae* and *E. coli* among the vegan and omnivore groups. **B**: Abundances of *Enterobacteriaceae* and *E. coli* among composite samples within the vegan and omnivore groups. TSS: total sum scaling, i.e., normalized proportional abundance. O1–O5 represent omnivores, and V1-V5 represent vegans (*p* < 0.001).

**Figure 2 microorganisms-08-01165-f002:**
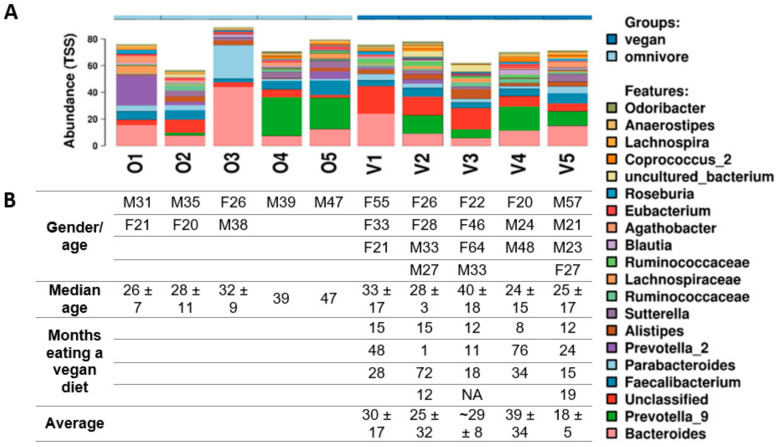
**A**: Genus level diversity analysis of the top 20 genera using 1000 operational taxonomic units (OTUs) in vegan and omnivore composite samples. Composite samples were composed of one to four samples. O1-O5 represent omnivores, and V1-V5 represent vegans. TSS: total sum scaling, i.e., normalized proportional abundance. **B**: Participant details (gender followed by their age, e.g., M31 indicates a 31- year-old male), median age ± SD. Months (and average) of vegan diets for each individual are presented under their diversity profiles.

**Figure 3 microorganisms-08-01165-f003:**
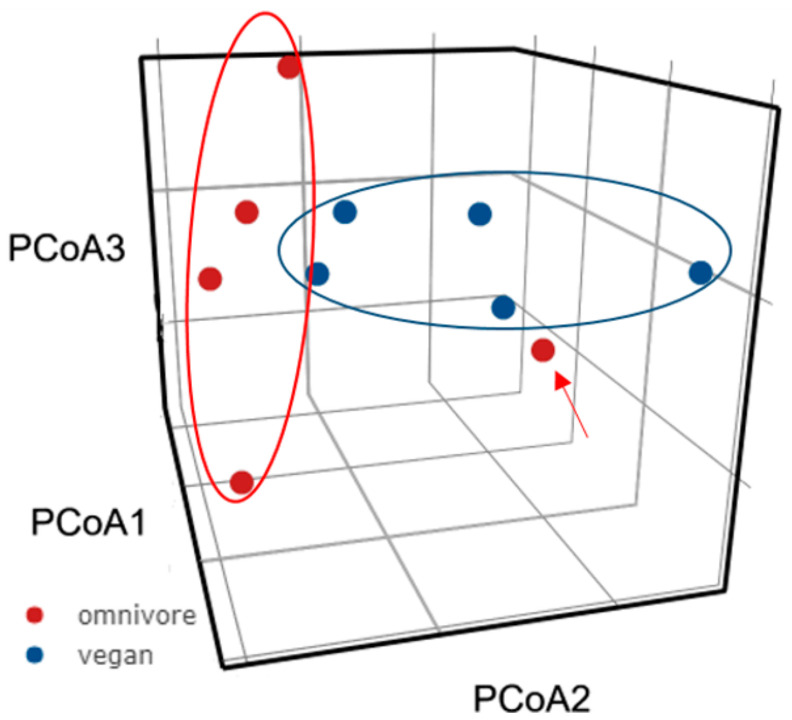
Principal coordinate analysis (PCoA) of microbiota similarity distribution of vegans and omnivores.

**Figure 4 microorganisms-08-01165-f004:**
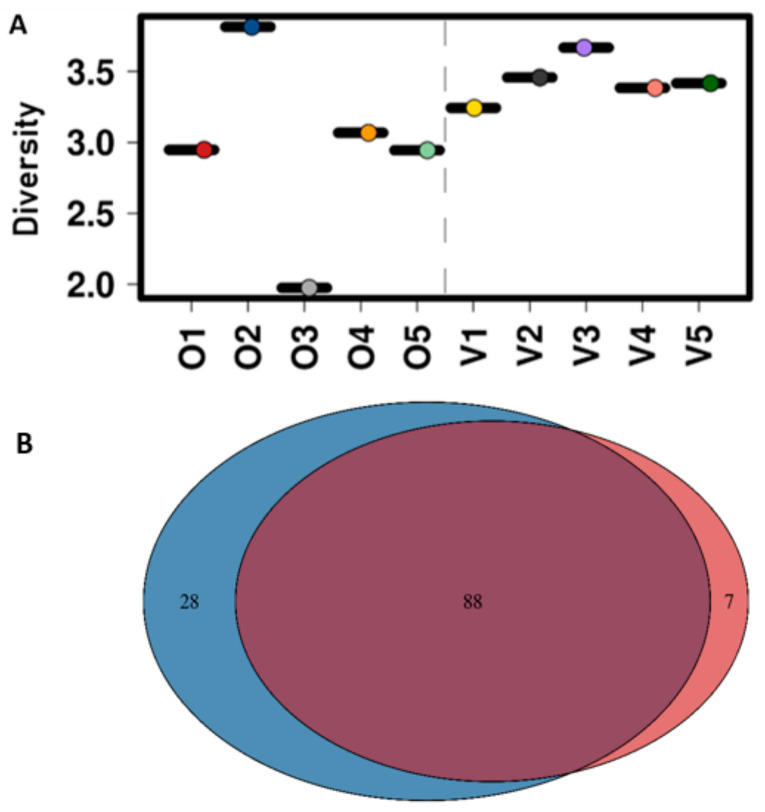
**A**: Overall genus level of microbial diversity using Shannon diversity in vegan and omnivore composite samples. O1-O5 represent omnivores, and V1-V5 represent vegans. **B**: Venn diagram representing the unique and shared microbial taxa among the vegan and omnivore groups. Blue represents vegans, purple represents shared taxa, and red represents omnivores.

**Table 1 microorganisms-08-01165-t001:** Demographic information about the participants in this study.

Participant Diet	Participant No.	Female	Male	Age Range (Years)	Median Age (Years)	Months Being Vegan	Mean± SD Months Being Vegan
**Vegan**	34	22	12	20–67	29 ± 1	1–108	36 ± 29
**Omnivore**	27	14	13	20–60	29 ± 1	NA	NA

**Table 2 microorganisms-08-01165-t002:** The number of fecal *E. coli* isolates (mean ± SEM) and their diversity in vegans and omnivores. Diversity was calculated as Simpson index of diversity.

Gender	*E. coli* CFU/g of Faeces		
	Vegan	Omnivore	*p*-Value
**Female**	9.3 ± 4.4 × 10^5^	7.9 ± 3.8 × 10^5^	0.602
**Male**	9.7 ± 6.2 × 10^6^	5.9 ± 3.1 × 10^5^	0.079
**Combined Gender**	3.4 ± 2.2 × 10^6^	9.6 ± 2.4 × 10^5^	0.116
	***E. coli* Diversity**		
**Female**	0.195 ± 0.06	0.199 ± 0.06	0.608
**Male**	0.201 ± 0.09	0.112 ± 0.06	0.519
**Combined Gender**	0.197 ± 0.05	0.157 ± 0.04	0.879

**Table 3 microorganisms-08-01165-t003:** The number of unique common (CT) and single types (ST) of *E. coli* found among vegans (*n* = 34) and 27 omnivores (*n* = 27) (pp = per participant).

	No. of CTs	Average CTs pp	CT Range pp	Total STs	No. of STs	Average STs pp	ST Range pp
**Vegan**	59	1.7	1–5	48	30	0.9	0–8
**Omnivore**	40	1.5	1–4		18	0.7	0–6

**Table 4 microorganisms-08-01165-t004:** Presence of AIEC-associated virulence genes (VGs) among 99 common types (CT) found in vegan and omnivore groups. The occurrence (as percentage) of each VG was calculated considering the number of *E. coli* within the identified CTs over the total number of *E. coli*.

	No. of CTs		No. of *E. coli* within CTs		Occurrence	
Virulence Genes	Vegan *n* = 59	Omnivore *n* = 40	Vegan *n* = 922	Omnivore *n* = 738	Vegan	Omnivore
***htr*** **A**	56	40	901	738	97%	100%
***lpf*** **A**	25	18	500	316	54%	43%
***omp*** **C**	59	40	922	738	100%	100%
***clb*** **A**	7	6	77	113	8%	15%
***dsb*** **A**	48	34	853	633	93%	86%
***afa*** **C**	17	18	338	287	37%	39%
***afa*** **C and *lpf*A**	11	14	241	213	26%	29%
***afa*** **C and *clb*A**	1	0	8	0	1%	0%
***afa*** **C and *dsb*A**	17	17	338	283	37%	38%
***dsb*** **A and *htr*A**	48	34	853	633	93%	86%

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
