# Peer review of "A Comparative Study of the Adherent-Invasive Escherichia coli Population and Gut Microbiota of Healthy Vegans versus Omnivores"

_microorganisms, 2020, doi:10.3390/microorganisms8081165_

Round 1
Reviewer 1 Report
In this research article the authors investigated microbiota variations among vegans and omnivores. Additionally, they evaluated the presence of adherent-invasive Escherichia coli (AIEC), a pathogen that is associated with inflammatory bowel disease (IBD), in the feces of the participants.
The highlight of this text is that the authors investigated microbiota variations that are linked to dietary habits in local populations.
Major points:
Firstly, the number of participants is rather small, a fact that is addressed by the authors, limiting the statistical power of the results. Additionally, several confounding factors, such as the specific diet, age, and gender are not considered. Also, the authors have grouped several samples in the vegan and omnivore groups, for logistical reasons, however, it is not clear on which basis these samples were grouped together.
Secondly, the authors refer at length to the fact that AIEC is correlated with IBD, however, as the study did not recruit IBD patients, this reference could be misleading. Assumptions cannot be made on the possible beneficial effects of diet in IBD based on the effect of diet on healthy individuals. The authors should keep these references should AIEC plays a role in the development of IBD. In that sense, differences in AIEC presence in the feces of healthy vegans or omnivores could be utilized for the investigation of the possible preventative actions of specific diets against the establishment of AIEC populations and IBD onset. However, if such a relationship does not exist, then the experimental design is not suited for such assumptions. The establishment of AIEC populations in the gut of diseased individuals can be determined by several other factors other than the diet (e.g. inflammation status), that should be also taken into consideration as confounding factors in future studies.
Additionally, the effect of vegan diets on the gut microbiota composition has been examined a number of times (Ferrocino et al., 2015, Losasso et al., 2018, Wu et al., 2016, Zimmer et al., 2012), so that this paper does not offer new insights, other than the effect of geographic location on gut microbiota composition in combination with different dietetic preferences.
Other points that require attention throughout the text, are:
- Lines 37-39: the authors should provide additional information on the correlation of AIEC with IBD and diet.
- Lines 80-95: Were there any health assessments conducted prior to the recruitment of the volunteers?
- Lines 97-103: Was there any means of ensuring the viability of the strictly anaerobic gut commensals? How was the possibility of contamination (e.g. skin microbiota) eliminated?
- Line 154: the omnivore samples presented in Figure 2B are 8 and not 18, appropriate corrections should be made
- Lines 237-238: the supplementary materials do not contain supplementary figure 2
- Line 358: the authors should add the number of the figure that they are referring to
References:
Ferrocino I, Di Cagno R, De Angelis M, et al. Fecal Microbiota in Healthy Subjects Following Omnivore, Vegetarian and Vegan Diets: Culturable Populations and rRNA DGGE Profiling. PLoS One. 2015;10(6):e0128669. Published 2015 Jun 2. doi:10.1371/journal.pone.0128669
Losasso C, Eckert EM, Mastrorilli E, et al. Assessing the Influence of Vegan, Vegetarian and Omnivore Oriented Westernized Dietary Styles on Human Gut Microbiota: A Cross Sectional Study. Front Microbiol. 2018;9:317. Published 2018 Mar 5. doi:10.3389/fmicb.2018.00317
Wu GD, Compher C, Chen EZ, et al. Comparative metabolomics in vegans and omnivores reveal constraints on diet-dependent gut microbiota metabolite production. Gut. 2016;65(1):63-72. doi:10.1136/gutjnl-2014-308209
Zimmer J, Lange B, Frick JS, et al. A vegan or vegetarian diet substantially alters the human colonic faecal microbiota. Eur J Clin Nutr. 2012;66(1):53-60. doi:10.1038/ejcn.2011.141
Author Response
Reviewer 1
In this research article the authors investigated microbiota variations among vegans and omnivores. Additionally, they evaluated the presence of adherent-invasive Escherichia coli (AIEC), a pathogen that is associated with inflammatory bowel disease (IBD), in the feces of the participants.
The highlight of this text is that the authors investigated microbiota variations that are linked to dietary habits in local populations.
Major points:
Firstly, the number of participants is rather small, a fact that is addressed by the authors, limiting the statistical power of the results. Additionally, several confounding factors, such as the specific diet, age, and gender are not considered. Also, the authors have grouped several samples in the vegan and omnivore groups, for logistical reasons, however, it is not clear on which basis these samples were grouped together.
Our response:
We agree with the reviewer that the number of participants is rather small and as noted by the reviewer, we have addressed this issue. The logistical reason mentioned in our manuscript for preparing 10 pooled samples (5 from vegans and 5 from omnivores) was that at the time of collecting samples, we were going to send a bigger DNA collection from another study to the US for 16S rDNA sequencing. To make up a total of 120 samples for a reduced cost, although we have not collected all our vegan and omnivore samples, we decided to include all 26 samples collected at the time in that shipment. Of course, we made sure that we would have 5 composite samples for each group.
We also agree with the reviewer that factors such as specific diet, age, and gender are not considered in this study although such information are provided in Figure 1B, but please note that the specific aim of the study was to compare the population structure of adhering-invasive E. coli between these two groups, where up to 28 colonies from each participant (>1700 isolates in total) were tested providing a greater than 90 % chance of detecting minor clones amongst the host E. coli faecal population as suggested by Schlager et al., 2002 Clonal diversity of Escherichia coli colonizing stools and urinary tracts of young girls. Infect Immun 70, 1225–1229.
Secondly, the authors refer at length to the fact that AIEC is correlated with IBD, however, as the study did not recruit IBD patients, this reference could be misleading. Assumptions cannot be made on the possible beneficial effects of diet in IBD based on the effect of diet on healthy individuals. The authors should keep these references should AIEC plays a role in the development of IBD. In that sense, differences in AIEC presence in the feces of healthy vegans or omnivores could be utilized for the investigation of the possible preventative actions of specific diets against the establishment of AIEC populations and IBD onset. However, if such a relationship does not exist, then the experimental design is not suited for such assumptions. The establishment of AIEC populations in the gut of diseased individuals can be determined by several other factors other than the diet (e.g. inflammation status), that should be also taken into consideration as confounding factors in future studies.
Our response:
Our initial hypothesis stemmed from the findings of many studies that have shown certain E. coli strains are closely associated with the mucosal membrane of IBD patients. These strains are shown to carry specific virulence genes not commonly found among commensal E. coli strains and named as AIEC. In this study we did not aim to establish that AIEC are correlated with IBD but since the diet is shown to influence gut inflammation and increases the risk of IBD, we aimed to investigate the diversity and prevalence of AIEC as well as the diversity profile of gut microbiota in healthy individuals consuming a vegan versus an omnivore diet. Special attention was given to the presence of AIEC strains in these two groups. We agree with the reviewer that the AIEC populations in the gut of diseased individuals can be due to several factors other than the diet but in this study, we focused on healthy individuals as indicated in the Materials and Methods (see lines 91-93).
Additionally, the effect of vegan diets on the gut microbiota composition has been examined a number of times (Ferrocino et al., 2015, Losasso et al., 2018, Wu et al., 2016, Zimmer et al., 2012), so that this paper does not offer new insights, other than the effect of geographic location on gut microbiota composition in combination with different dietetic preferences.
Our response:
The reviewer is correct that the effect of vegan diets on the gut microbiota composition has been examined by others. We certainly aware of that and we have referenced them as well as mentioned by the reviewer (please see 14, 18, 19 and 20). However, to the best of our knowledge no study has investigated the prevalence and diversity of AIEC in vegans compared to omnivores.
Other points that require attention throughout the text, are:
- Lines 37-39: the authors should provide additional information on the correlation of AIEC with IBD and diet.
Our response
Whist there are many studies correlating the diet and IBD or AIEC and IBD, we could not find any study to correlate IBD, diet and AIEC. There are, however, several studies that report the relation between gut microbiota and diet in which overgrowth of E. coli population due to high fat and high sugar diet is correlated with colitis. In the discussion section we have also pointed at the changes in the E. coli populations in vegans or vegetarians (see lines 297-300).
Lines 80-95: Were there any health assessments conducted prior to the recruitment of the volunteers?
Our response
We recruited participants via survey, face-to face invitation and their self-assessment. All participants confirmed that they had not taken antibiotics or probiotics within the past 3 months and had no health conditions associated with the gastrointestinal tract and therefore were considered as apparently healthy (see lines 88-94)
- Lines 97-103: Was there any means of ensuring the viability of the strictly anaerobic gut commensals? How was the possibility of contamination (e.g. skin microbiota) eliminated?
Our response
For measuring the diversity profile of the gut microbiota, we extracted DNA of the whole microbial genome of the faecal samples. This provided us with the DNA of entire microbiota (live or dead) and we used that for 16S rDNA analyses. Participants were provided with a sampling package which contained sterile plastic gloves for sampling (to avoid direct contact of faeces with the hand of the participant), disposable faeces catcher, disposable face mask , sterile faecal specimen container, transport media and sterile swab and sealed bag. The package also contained a full instruction as how to collect faces and store them before they were collected by a member of the team (see lines 102-104).
Line 154: the omnivore samples presented in Figure 2B are 8 and not 18, appropriate corrections should be made
Correction made (line 156).
- Lines 237-238: the supplementary materials do not contain supplementary figure 2
We apologise for the mistake. It should be supplementary figure 1. See line 241 in the revised manuscript
- Line 358: the authors should add the number of the figure that they are referring to
Correction made (see line 361-revised manuscript)
Reviewer 2 Report
Veca et al. present a paper in which they aimed to study the population of E. coli and gut microbiota as a whole in vegans vs. omnivores. It is an interesting study, yet I consider there are some important issues that need to be addressed. Furthermore, on a whole, the paper should be carefully revised regarding fluidity in order to improve its readability, which at times is difficult and impedes understanding what the authors are trying to convey.
Major comments:
- The fact that the authors are studying gut microbiota in two different populations characterized by their diet (i.e. vegans vs. omnivores) and there is no recording of the dietary intake of these participants is quite problematic. There is no way that one can make any type of assumptions if there is no data regarding fiber intake, protein, insoluble carbohydrates, etc., all known to majorly affect gut microbiota populations.
- The n number is quite low compared to similar studies, which is probably why there are no visible significant differences between groups.
- Due to the somewhat confusing writing at times, the aim is not very clear, as well as the study conclusions. Why did the authors consider this important to study? What does this bring to current literature? What are future steps?
Other comments:
Title: The title contains E.coli twice, I suggest rephrasing.
Abstract
Lines 14-15: I suggest adding “We assessed the influence of vegan and omnivore diets on human gut microbiota composition, and the prevalence of AIEC, in healthy adults to determine/with the aim to XXXX” (or similar, the aim should be clearer).
Introduction
The fact this study is done in healthy individuals strikes with the emphasis you give on IBD and others, which are mentioned in both the abstract and the first paragraph of the introduction, where you discuss the health and economic burden of these conditions. Yet, you do not carry out the study in this population. I suggest rephrasing and rewriting these sections so that it is clear to the reader that you are studying a bacteria/gut microbiota in healthy individuals.
Materials and methods
- In table 1, it says that the range of months of being vegan was 1-108, I think you should indicate the n number for each range or group (i.e. 1-10 months, 11-20 or as you consider appropriate). It gives the reader a better sense of the population studied.
- How were participants analysed regarding their compliance to diet?
- For the analysis of fecal samples, you say that you pooled 18 samples, however the n is higher in each group. Why did you only use 18? Why not use the rest?
- Why is the n number so low? Compared to many similar studies, it seems to be very low. I think it would be interesting to discuss this.
Results:
- Figure 2B is quite confusing, is there a way to simplify it?
Discussion
- In line 293, you mention a study by Zimmer et al 2012 and reference it as ref. nº 41, yet it seems to be 14. Please revise all you references to ensure they are correct.
- Line 312: you did not assess the impact of the gut microbiota and e coli populations, as far as I understand, on the participants. Please rephrase.
- Line 314: statically should be statistically
Examples of sentences that should be rephrased/revised (although it is not an exhaustive list, just examples):
-lines 33-35
Line 60: vegan vs. an omnivorous diet
Line 94: Study was approved by human ethics, University of the Sunshine Coast (S/17/1122). This sentence is written incorrectly.
Line 99: Containers containing…
Line 150: Known quantity of fecal samples…
Line 236: For logistical reasons, this data was visualized using the top 20 genera and 1000 OTUs, Supplementary Figure 2 displays this data with the top 100 genera using 5000 OTUs for comparison.
The conclusion is also quite diffuse.
Author Response
Reviewer 2
Veca et al. present a paper in which they aimed to study the population of E. coli and gut microbiota as a whole in vegans vs. omnivores. It is an interesting study, yet I consider there are some important issues that need to be addressed. Furthermore, on a whole, the paper should be carefully revised regarding fluidity in order to improve its readability, which at times is difficult and impedes understanding what the authors are trying to convey.
Major comments:
The fact that the authors are studying gut microbiota in two different populations characterized by their diet (i.e. vegans vs. omnivores) and there is no recording of the dietary intake of these participants is quite problematic. There is no way that one can make any type of assumptions if there is no data regarding fiber intake, protein, insoluble carbohydrates, etc., all known to majorly affect gut microbiota populations.
Our response:
We agree with the reviewer that studies measuring the impact of nutritional diet on gut microbiota should consider factors such as fibre intake, protein, insoluble carbohydrates etc. as indicated by the reviewer. However, the main aim of our study was to compare the population structure of AIEC strains in vegans and omnivores and therefore no specific nutritional components in these groups were recorded. In contrast we defined participants as vegan if they did not consume meat, poultry, fish, dairy, eggs and gelatine for a minimum of 4 weeks while omnivores were defined those that included all the above products in their diets (see lines 83-88). Except for dietary exclusions in vegans, the participants in this study were not required to present a food journal or follow any strict dietary guidelines within the designated diet groups.
The n number is quite low compared to similar studies, which is probably why there are no visible significant differences between groups.
Our response:
We agree with the reviewer and have addressed this limitation in the discussion. However because the main scope of this study was to assess the prevalence of AIEC population, in these groups we tried to strengthen our work by testing 28 colonies of E. coli from each participant (>1700 E. coli isolates in total). This allowed us to detect different clones of E. coli in each participant with more than 90 % chance of detecting minor clones amongst the host E. coli faecal population (see: Schlager et al., 2002 Clonal diversity of Escherichia coli colonizing stools and urinary tracts of young girls. Infect Immun 70, 1225–1229).
Due to the somewhat confusing writing at times, the aim is not very clear, as well as the study conclusions. Why did the authors consider this important to study? What does this bring to current literature? What are future steps?
Our response:
In the revised version we have tried to improve the quality of writing to clarify aims and the importance of the study in the abstract (lines 13-16) and throughout the manuscript where necessary including. We considered undertaking this study for two main reasons. The first reason was that many studies investigating gut microbiota and E. coli populations in IBD patients have shown that certain E. coli strains have been closely associated with the mucosal membrane of IBD patients. These strains were shown to carry specific virulence genes that were not commonly found among commensal E. coli strains. Due to their specific virulence characteristics and interaction with gut epithelial cell lines they were named as AIEC. The second reason which also prompted us to carry this study was the fact that literatures show a strong correlation between diet with the IBD. In view of the above we hypothesised that AIEC may be more commonly associated with what we defined as western diet, or omnivores, as compared with vegans. The importance of such study, which we intend to continue with analysing more participants and an in-depth analyses of gut microbiota/AIEC, would be to identity individuals at risk of developing IBD, if their E. coli population carry AIEC genes
Other comments:
Title: The title contains E. coli twice, I suggest rephrasing.
Our response:
We have re-phrased the title to “A comparative study of the adherent-invasive Escherichia coli population and gut microbiota of healthy vegans versus omnivores” (Lines 2-4)
Abstract
Lines 14-15: I suggest adding “We assessed the influence of vegan and omnivore diets on human gut microbiota composition, and the prevalence of AIEC, in healthy adults to determine/with the aim to XXXX” (or similar, the aim should be clearer).
Our response:
Thanks for reviewer’s suggestion, we rephrased the Abstract to make the aim of the study clear (see lines 12-16).
Introduction
The fact this study is done in healthy individuals strikes with the emphasis you give on IBD and others, which are mentioned in both the abstract and the first paragraph of the introduction, where you discuss the health and economic burden of these conditions. Yet, you do not carry out the study in this population. I suggest rephrasing and rewriting these sections so that it is clear to the reader that you are studying a bacteria/gut microbiota in healthy individuals.
Our response:
The first few lines of the abstract and the introduction we aimed to highlight the importance of the IBD globally and to introduce the findings of recent studies on the presence of a specific group of E. coli, termed as AIEC, in IBD patients as a possible bacterial aetiology for the disease (see lines 36-42). We believe that it would be important for the reader to get a clear message on the role of AIEC in the pathogenesis of IBD and the presence of AIEC in gut of healthy individuals as a risk factor for the development of IBD in future. Of course we make it clear that to investigate the impact of a vegan diet on carriage of AIEC, this study well compare the prevalence of AIEC between two groups of healthy vegans and omnivores (See aim of the study lines 76-79).
Materials and methods
In table 1, it says that the range of months of being vegan was 1-108, I think you should indicate the n number for each range or group (i.e. 1-10 months, 11-20 or as you consider appropriate). It gives the reader a better sense of the population studied.
Our response:
We divided our vegetarian participants into 4 groups based on the months of being vegetarian as recommended by the reviewer i.e. 1-10, 11-20, 21-30 and >30 months (lines 95-97). We also provided a mean of months of being vegetarian ± SD in table 1 to give a better understanding of the vegan population (see revised Table 1)
How were participants analysed regarding their compliance to diet?
Our response:
This was done when we recruited participants which was done via survey, face-to-face meeting and participant’s self-assessment when completing the questionnaire (see line 89).
For the analysis of fecal samples, you say that you pooled 18 samples, however the n is higher in each group. Why did you only use 18? Why not use the rest?
Our response:
We apology for the confusion made in giving the right number. In all 26 faecal samples were sent for analysis of faecal samples via 16S rRNA analyses. This included 18 vegan samples and 8 omnivore samples as also shown in Figure 2. This was made clear in the text (lines 156-158). These sample were pooled as 10 composite samples (5 from vegans and 5 from omnivores).
Why is the n number so low? Compared to many similar studies, it seems to be very low. I think it would be interesting to discuss this.
Our response:
We agree with the reviewer that the number of samples was low and we have already addressed this in the discussion. On the other hand because the main goal of the study was to compare the population structure of AIEC between the vegans and omnivores, we made sure that we collect and analyse a high number of E. coli isolates from the faeces of each individual i.e. 28 E. coli isolates from each participant (>1700 isolates in total).
Results:
Figure 2B is quite confusing, is there a way to simplify it?
Our response:
We aligned the figure 2a showing the diversity profile of gut microbiota with data that show demographic information of participants from whom faecal samples were analysed for diversity of the gut microbiota (figure 2b). The alignment was carefully made to see all demographic information locate right underneath each column of the graph in fig 2a.
Discussion
In line 293, you mention a study by Zimmer et al 2012 and reference it as ref. nº 41, yet it seems to be 14. Please revise all you references to ensure they are correct.
Our response:
The references on this part have been checked and edited throughout (see lines 296 and 301).
Line 312: you did not assess the impact of the gut microbiota and e coli populations, as far as I understand, on the participants. Please rephrase.
Our response:
We rephrased this sentence to avoid confusion (see lines 316-317).
Line 314: statically should be statistically
Our response:
Thanks for identifying this error. This has been corrected in the text (Line 318).
Examples of sentences that should be rephrased/revised (although it is not an exhaustive list, just examples):
-lines 33-35
Our response:
We removed ‘presenting’ and added ‘with’ as recommended (see line 35).
Line 60: vegan vs. an omnivorous diet
Our response:
Correction has been made in text (line 61).
Line 94: Study was approved by human ethics, University of the Sunshine Coast (S/17/1122). This sentence is written incorrectly.
Our response:
We have reworded this sentence to: “Human ethics for this study was approved by the University of the Sunshine Coast (S/17/1122).” (Line 98-99).
Line 99: Containers containing…
Our response:
The sentence was rephrased (see lines Line 103-105 of the revised version).
Line 150: Known quantity of fecal samples…
Our response:
This sentence was rephrased for clarity (see line 154-155).
Line 236: For logistical reasons, this data was visualized using the top 20 genera and 1000 OTUs, Supplementary Figure 2 displays this data with the top 100 genera using 5000 OTUs for comparison.
Our response:
In the revised version this was corrected as below: “Data was visualized using the top 20 genera and 1000 OTUs (Figure 2). For further information on the top 100 genera using 5000 OTUs see Supplementary Figure 1.” (See lines 240-242).
The conclusion is also quite diffuse.
Our response:
The conclusion part was rephrased to make it clear (see lines 348-353).
“In conclusion, the abundance and diversity of E. coli populations as well as the prevalence of AIEC-associated VGs did not differ between the two diet groups. Microbiota analysis revealed more unique taxa amongst the vegans compared to omnivores. Specifically, Bacteroides was found in higher proportion in the omnivore group, which could be due to the diet, whereas Enterobacteriaceae was higher in the vegan group. Further study is needed to identify the importance of these two groups in relation to the diet.”
Round 2
Reviewer 1 Report
The authors have addressed most of my comments.
However, I need to mention that their answer to my first comment (We agree with the reviewer that the number of participants is rather small and as noted by the reviewer, we have addressed this issue. The logistical reason mentioned in our manuscript for preparing 10 pooled samples (5 from vegans and 5 from omnivores) was that at the time of collecting samples, we were going to send a bigger DNA collection from another study to the US for 16S rDNA sequencing. To make up a total of 120 samples for a reduced cost, although we have not collected all our vegan and omnivore samples, we decided to include all 26 samples collected at the time in that shipment. Of course, we made sure that we would have 5 composite samples for each group) lacks scientific soundness.
Nevertheless, their manuscript has been improved and it is now suitable for publication.
Author Response
We thank the reviewer for his/her constructive comments and suggestions throughout reviewing our manuscript and for understanding our limitation with regard to sample size and logistics of analyzing the whole samples
regards
M. Katouli (corresponding author)
Reviewer 2 Report
Thank you for revising you mansucript and providing some of the changes suggested.
I still suggest you revise the English and writing style, since there are still many errors (including in the sentences you added in your revised version).
Examples include:
Abstract: Please remove brackets before “of healthy” and add before n=61:
- as well as the composition gut microbiolta in faecal samples (of healthy participants n=61) on a vegan (n=34) or omnivore (n=27) diets.
Introduction:
Line 35: should be health-associated
Line 45-47: Please revise, it’s hard to understand
Line 62: vurses should be versus
Conclusion:
Line 356: please revise, its incorrect
These are just examples; please revise throughout.
Author Response
We thank the reviewer for his/her constructive comments and for identifying areas that require correction. In this revised version, we made corrections where pointed out by the reviewer (high lighted in yellow). In particular the changes suggested to lines 42-54, which has been rephrased for greater clarity (the entire paragraph has been high lighted in yellow). The manuscript has also been checked for English writing style and areas that were unclear have been amended (high lighted in green).
Thanks
M. Katouli (corresponding author)